# Mathematical framework for place coding in the auditory system

**Alex D. Reyes** *

Center for Neural Science, New York University, New York, New York, United States of America

* ar65@nyu.edu

## Abstract

In the auditory system, tonotopy is postulated to be the substrate for a place code, where sound frequency is encoded by the location of the neurons that fire during the stimulus. Though conceptually simple, the computations that allow for the representation of intensity and complex sounds are poorly understood. Here, a mathematical framework is developed in order to define clearly the conditions that support a place code. To accommodate both frequency and intensity information, the neural network is described as a space with elements that represent individual neurons and clusters of neurons. A mapping is then constructed from acoustic space to neural space so that frequency and intensity are encoded, respectively, by the location and size of the clusters. Algebraic operations -addition and multiplication- are derived to elucidate the rules for representing, assembling, and modulating multi-frequency sound in networks. The resulting outcomes of these operations are consistent with network simulations as well as with electrophysiological and psychophysical data. The analyses show how both frequency and intensity can be encoded with a purely place code, without the need for rate or temporal coding schemes. The algebraic operations are used to describe loudness summation and suggest a mechanism for the critical band. The mathematical approach complements experimental and computational approaches and provides a foundation for interpreting data and constructing models.

## Author summary

One way of encoding sensory information in the brain is with a so-called place code. In the auditory system, tones of increasing frequencies activate sets of neurons at progressively different locations along an axis. The goal of this study is to elucidate the mathematical principles for representing tone frequency and intensity in neural networks. The rigorous, formal process ensures that the conditions for a place code and the associated computations are defined precisely. This mathematical approach offers new insights into experimental data and a framework for constructing network models.

**Data Availability Statement:** The code for the simulations can be downloaded at https://github.com/AlexDReyes/ReyesPlosComp.git.

**Funding:** The author received no specific funding for this work.

**Competing interests:** The authors have declared that no competing interests exist.

## Introduction

Many sensory systems are organized topographically so that adjacent neurons have small differences in the receptive fields. The result is that minute changes in the sensory features causes an incremental shift in the spatial distribution of active neurons. This is has led to the notion of a place code where the location of the active neurons provides information about sensory attributes. In the auditory system, the substrate for a place code is tonotopy, where the preferred frequency of each neuron varies systematically along one axis [1]. Tonotopy originates in the cochlea [2, 3] and is inherited by progressively higher order structures along the auditory pathway [4]. The importance of a place code [5] is underscored by the fact that cochlear implants, arguably the most successful brain-machine interface, enable deaf patients to discriminate tone pitch simply by delivering brief electrical pulses at points of the cochlea corresponding to specific frequencies [6, 7].

Although frequency and intensity may be encoded in several ways [8], there are regimes where place-coding seems advantageous. Humans are able to discriminate small differences in frequencies and intensities even for stimuli as brief as 5–10 ms [9–13]. Therefore, the major computations have already taken place within a few milliseconds. This is of some significance because in this short time interval, neurons can fire only bursts of 1–2 action potentials [14, 15], indicating that neurons essentially function as binary units. Therefore, it seems likely that neither frequency nor intensity can be encoded via the firing rate of individual cells since the dynamic range would be severely limited. Similarly, coding schemes based on temporal or 'volley' schemes are difficult to implement at the level of cortex because neurons can phase-lock only to low frequency sounds [16–18]. However, a purely place code cannot be used for dynamically complex sound; indeed, coding and perception are enhanced significantly when temporal and rate cues are factored in [8, 12, 19–22] and when longer duration stimuli are used [9–12].

There are several challenges with implementing a purely place coding scheme. First, the optimal architecture for representing frequency is not well-defined. Possible functional units include individual neurons, cortical columns [23, 24], or overlapping neuron clusters [25]. The dimension of each unit ultimately determines the range and resolution at which frequencies and intensities that can be represented and discriminated. Second, how both frequency and intensity can be encoded with a place code is unclear, particularly for brief stimuli when cells function mostly as binary units. Third, the rules for combining multiple stimuli is lacking. Physiological sounds are composed of pure tones with differing frequencies and intensities, resulting in potentially complex spatial activity patterns in networks. Finally, the role of inhibition in a place coding scheme has not been established.

Here, a mathematical model is developed in order to gain insights into: 1) the functional organization of the auditory system that supports a place coding scheme for frequency and intensity: and 2) the computations that can be performed in networks. To simplify analyses and to reveal the inherent advantages and limitations, the model focuses on how simple tones are represented and combined with a pure place code and excludes the dynamic variables that mediate temporally complex sounds. The approach is to use mathematical principles to construct the acoustic and neural spaces, find a mapping between the spaces, and then develop the algebraic operations. The predictions of the math model are then tested with simulations. With this formal approach, the variables that are important for a place coding scheme are defined precisely.

## Results

The mathematical model is subject to the following biological constraints. First, the neural network inherits the tonotopic organization of the cochlea [2, 3] so that the preferred frequency

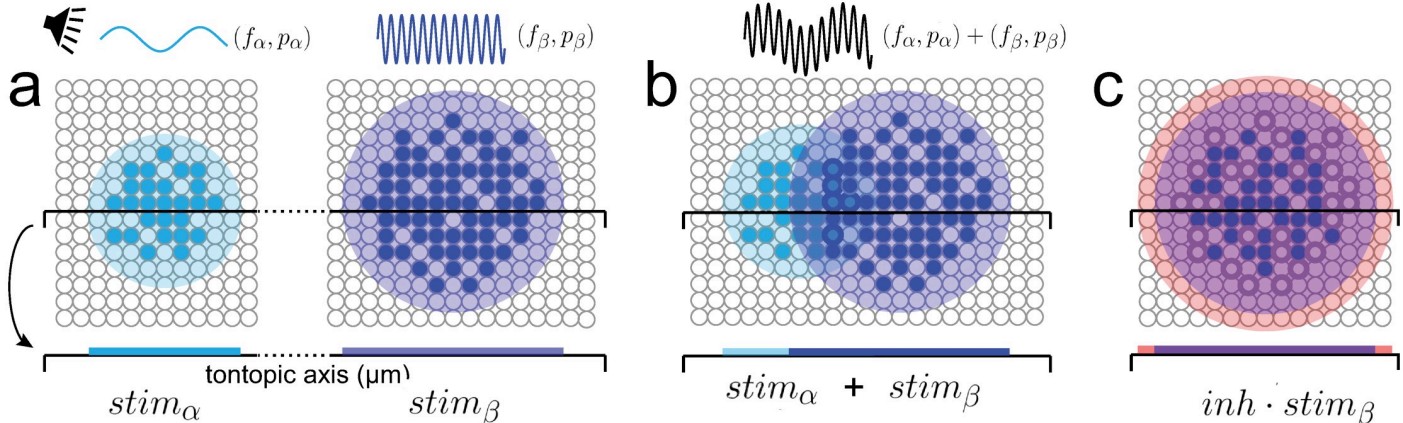

**Fig 1. Computations with a place code.** *a, left*, hypothetical neural representation of a low frequency ($f_\alpha$), small amplitude ($p_\alpha$) pure tone stimulus in a two-dimensional neural network. A stimulus-evoked synaptic field covers a circular area (cyan disk) and causes a subset of cells to fire (filled circles). Projection of the synaptic field onto the tonotopic axis (cyan bar) gives the location and size of the activated area. *right*, synaptic field generated by a tone with higher frequency ($f_\beta$) and sound pressure ($p_\beta$). *b*, synaptic field generated by a sound composed of the two tones. *c*, modulation by inhibition (red).

of neurons changes systematically with location along one axis. Second, a pure tone activates a population of neurons within a confined area [26], with the location of the area varying with the tone frequency. Third, the area grows with sound intensity [26], paralleling the increase in the response area of the basilar membrane [27, 28]. The model is broadly related to the "spread-of-excitation" class of models [8, 19]

The basic computations that can be performed using a place code are shown schematically in a 2-dimensional network of neurons (Fig 1). In response to a pure tone stimulus, a synaptic field from a presynaptic population of neurons is generated within an enclosed area of the network (a, left panel, cyan disk), causing a population of cells to fire (filled circles). A tone with a higher frequency and intensity activates a larger area at a different location (right panel). A sound composed of the two pure tones activates both regions simultaneously; the regions may overlap if the difference in frequencies is small (Fig 1B). Finally, excitatory synaptic fields and the activated neuron clusters are modulated by inhibition (Fig 1C). The mathematical basis for these computations is developed below. For clarity, only the main results are shown, with details in S1 Appendix.

## Neural space

Although the brain has three spatial dimensions, only one dimension -that corresponding to the tonotopic axis- is relevant for a place code. Thus, the circular synaptic field in Fig 1 is projected onto the tonotopic axis (bars). In the presence of sound, afferents from an external source generates a synaptic field that covers a contiguous subset of neurons. In the following, a mathematical description of the neural space will be developed that accommodates the neural elements and synaptic fields.

The neural space is defined as an interval, bounded by minimum and maximum values $x_{min}$ and $x_{max}$ (Fig 2B, magenta). The neural space is partitioned into $N_h$ sections that represent the projections of neurons to the tonotopic axis (Fig 1). For explanation purposes, the neural space is constructed from single row of neurons (Fig 2A) (see S1 Appendix for a more general definition with multiple layers of staggered neurons). The neural space and the partitions are 'half-open' intervals, which are closed ("[") on one end and open on the other (")"). This is convenient mathematically because there is no space between intervals, allowing for a formal

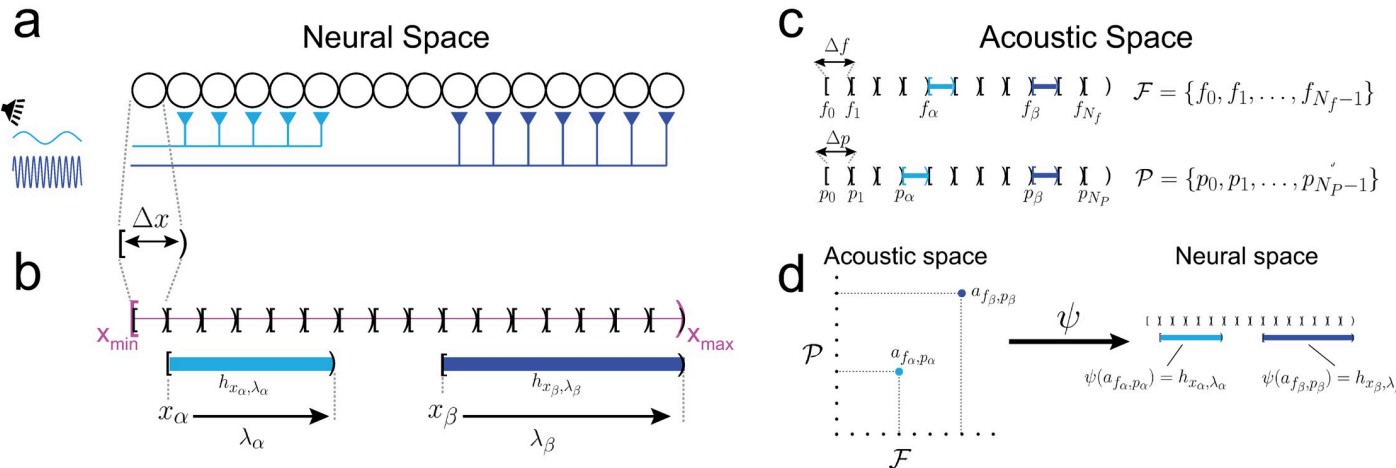

**Fig 2. Mathematical representation of neural and acoustic spaces.** *a*, neurons on the tonotopic axis are positioned next to each other with no space in between. Pure tone stimuli activate afferents onto a subset of neurons (cyan, blue). *b*, mathematical representation of neural space. The tonotopic axis is a half-open interval (magenta) partitioned into smaller intervals that represent the space ($\Delta x$) taken up by neurons. The synaptic fields ($h_{x_\alpha,\lambda_\alpha}$, $h_{x_\beta,\lambda_\beta}$) are also half-open intervals. *c*, The acoustic space has two dimensions, with frequency as one axis and pressure as the other. The frequency and pressure spaces are partitioned into half-open intervals of length $\Delta f$, $\Delta p$, respectively. *d*, Mapping tones in the acoustic space to intervals in neural space via a function $\psi$.

definition of a partition (S1 Appendix). Each interval may be expressed as non-overlapping subintervals of the form $[x, x + \Delta x)$. Therefore, $\Delta x$ may be viewed as the width of an individual neuron and $N_h$ as the number of cells (Fig 2B). Each interval can be uniquely identified by the point at the closed end, which also gives its location along the tonotopic axis. The set containing these points is:

$$\mathcal{X} = \{x_0, x_1, \ldots, x_{(N_h-1)}\} \tag{1}$$

A synaptic field spans an integral number ($n_\lambda$) of neurons and is also defined as a half-open interval with length $\lambda = n_\lambda \Delta x$. The length $\lambda$ ranges from $\Delta x$ (1 interval) to a maximum $\lambda_{max} = n_{\lambda,max}\Delta x$. Each synaptic interval, designated as $h_{x,\lambda} = [x, x + \lambda)$, is uniquely identified by the location of the cell at the closed end ($x_\alpha$, $x_\beta$ in Fig 2B) and by its length ($\lambda_\alpha$, $\lambda_\beta$). The set of starting points $\mathcal{X}^\lambda$ and the set of achievable lengths $\Lambda$ are given by:

$$\mathcal{X}^\lambda = \{x_i \mid i \in \mathbb{N}_0, i \le (N_h - n_{\lambda,max})\} \subset \mathcal{X}$$
$$\Lambda = \{\lambda_1, \lambda_2, \ldots, \lambda_{max}\} \tag{2}$$

The set $X^\lambda \subset \mathcal{X}$ takes into account the maximum interval length to ensure that the synaptic intervals are within neural space. A more formal and general definition of neural space and its topology is in S1 Appendix. As will be shown below, $\mathcal{X}^\lambda$ will contain information about the frequency while $\Lambda$ will contain information about the sound pressure.

## Representing sound in neural space

The elements of acoustic space $\mathcal{A}$ are pure tones, each of which is characterized by a sine wave of a given frequency ($f$) and amplitude corresponding to sound pressure ($p$). Theoretically, frequencies and pressures are unbounded and can take an uncountable number of values but under physiological conditions, the audible range is likely bounded by minimum and maximum values and consists of a finite number of discriminable frequencies and pressure levels.

The acoustic space has two dimensions with frequency as one axis and sound pressure as the other. As was done for neural space, the frequency and pressure axes are defined as a half-open intervals and divided into non-overlapping subintervals expressed as $[f, f + \Delta f)$ and $[p, p + \Delta p)$, respectively (Fig 2C). For example, two tones with frequencies $f_\alpha, f_\beta$ that are in the interval $[f_1, f_1 + \Delta f)$ will both be 'assigned' to $f_1$ (S1 Appendix). Physiologically, $\Delta f$ and $\Delta p$ limit the resolutions of frequency and sound pressure perception and set the lower bounds of difference limens (see Discussion). The sets of $N_f$ audible frequencies and $N_p$ pressures are:

$$\mathcal{F} = \{f_0, f_1, \ldots, f_{(N_f - 1)}\}$$
$$\mathcal{P} = \{p_0, p_1, \ldots, p_{(N_p - 1)}\} \qquad (3)$$

where the elements are the first points of each interval.

The number of audible frequencies and pressures are limited by the number of synaptic intervals that fit into the tonotopic axis ($|\mathcal{X}^\lambda|$) and the number of cells that fit into a single synaptic interval ($|\Lambda|$), respectively. For simplicity, $\mathcal{F}$ and $\mathcal{P}$ are on a linear, rather than logarithmic scales.

Single tones in acoustic space are represented in neural space via a mapping $\psi$ (Fig 2D; see S1 Appendix for formal treatment). A pure tone $a_{f,p} \in \mathcal{A}$ is mapped to an interval $h_{x,\lambda}$ by first mapping the components $f$ to $x$ and $p$ to $\lambda$ via $\psi_f(f) = x$ and $\psi_p(p) = \lambda$.

$$\psi(a_{f,p}) = h_{\psi_f(f), \psi_p(p)} = h_{x,\lambda} \qquad (4)$$

By adjusting $|\mathcal{F}|$ and $|\mathcal{P}|$ to match $|\mathcal{X}^\lambda|$ and $|\Lambda|$, respectively, the mapping of single tones onto intervals can be made bijective (one-to-one, onto).

A mapping from acoustic space to intervals that are inhibitory can be similarly defined. The mapping is complicated by the fact that there fewer inhibitory (**I**) cells than excitatory (**E**) [29], may have different tuning properties [30, 31], and can be in the co-tuned or lateral inhibitory configuration depending on the stimulus [32]. The mapping is under ongoing investigation but for purposes of the present analyses, the mapping is taken to be identical to that for **E**.

## Algebraic operations with synaptic intervals

Having formally described the mathematical structure of neural space, it is now possible to define the algebraic operations -addition and multiplication- for combining and modulating synaptic intervals (Fig 1B and 1C). To simplify notation, the intervals will henceforth be identified with a single subscript or superscript.

'Addition' of synaptic intervals is defined as their union. Let $h_\alpha, h_\beta$ be two synaptic intervals. Then,

$$h_\alpha + h_\beta \overset{\text{def}}{=} h_\alpha \cup h_\beta \qquad (5)$$

The addition operation yields two possible results (S1 Appendix). If the two synaptic intervals do not overlap ($h_\beta, h_\gamma$ in Fig 3A*i*), the union yields a set with the same two intervals (Fig 3A*ii*, bottom). If the two intervals overlap ($h_\alpha$ and $h_\beta$, $h_\alpha$ and $h_\gamma$), the result is a single interval whose length depends on the amount of overlap (Fig 3A*ii*, top two traces). Note that summation is sublinear: $|h_\alpha + h_\gamma| < |h_\alpha| + |h_\gamma|$. Moreover, if the starting points are the same ($h_\alpha$ and $h_\beta$), the length is equal to that of longer summand ($|h_\alpha + h_\beta| = |h_\alpha|, |h_\alpha| > |h_\beta|$).

It is noteworthy that there is some ambiguity from a decoding perspective if there are multiple tones because the addition operation will fuse overlapping intervals into a larger, single interval (e.g. $h_\alpha + h_\gamma$ in Fig 3A). It would not be possible to determine whether a synaptic

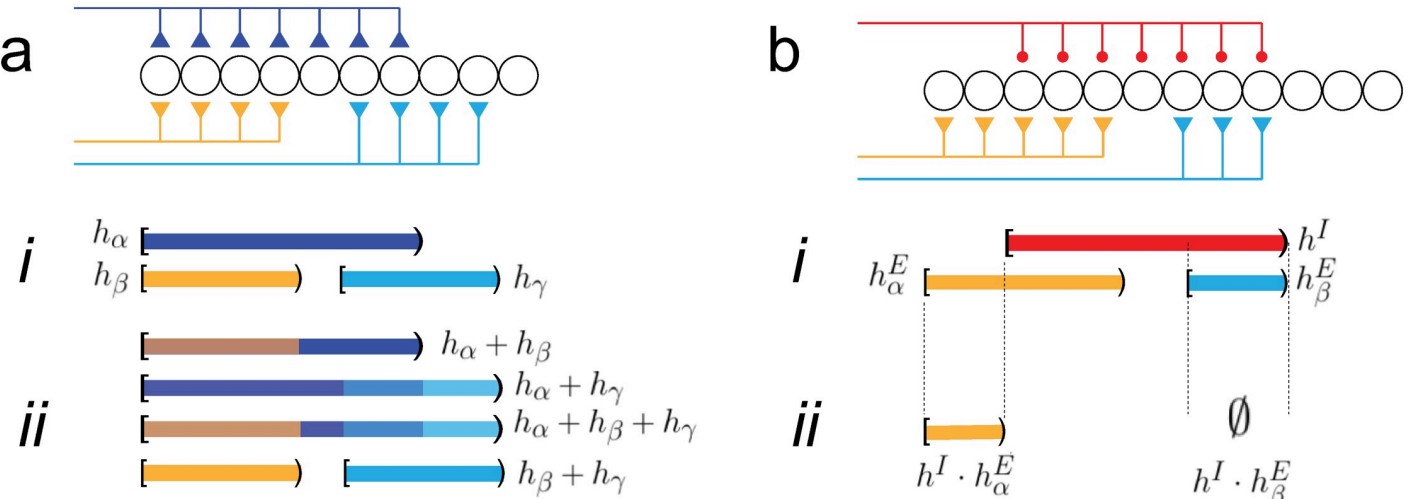

**Fig 3. Addition and multiplication.** *a*, schematic of network receiving three excitatory afferent inputs. *i*, half-open intervals representing the synaptic fields. *ii*, addition (union) of different combinations of intervals. *b*, schematic of network receiving two excitatory inputs (orange, cyan) and an inhibitory input (red). *i*, half-open intervals associated with the activated afferents. *ii*, multiplication (set minus) of each excitatory intervals by the inhibitory interval.

interval is a result of a single high intensity pure tone, multiple low intensity pure tones with small differences in frequencies, or band limited noise. There is some evidence of this ambiguity in psychophysical experiments (see Discussion).

One example of addition that occurs under biological conditions is when a pure tone arrives simultaneously to the two ears. The signal propagates separately through the auditory pathway but eventually converges at some brain region. Because each input is due to the same tone, the resultant synaptic intervals will be at same location (i.e. have the same starting points) on the tonotopic axis, though their lengths may differ because of interaural intensity differences (orange and blue intervals in Fig 3A). Fig 4A (top panel) shows the predicted total length when two intervals with different lengths are added (the length of the cyan interval is fixed while that of the blue is increased). The total length is equal to the length of the longer interval: hence, it is initially constant and equal to that of the cyan interval but then increases linearly when the length of the blue interval becomes longer.

Addition also takes place when the sound is composed of two pure tones with different frequencies. This would generate two synaptic intervals with different starting points and possibly different lengths (e.g. orange and cyan intervals in Fig 3A). The total length depends on the degree of overlap between the intervals. In Fig 4A (bottom panel), the location of one interval (cyan) is fixed while the other (blue) is shifted rightward. When the two intervals completely overlap, the total length is equal to the length of one interval. As the blue interval is shifted, the total length increases linearly and plateaus when the two intervals become disjoint.

Inhibition decreases the excitability of the network and would be expected to reduce the size of the synaptic interval. This is not possible with the addition operation because the union operation has no inverse (i.e. 'subtraction' is not defined; S1 Appendix). Therefore, to incorporate the effects of inhibition, a 'multiplication' operation is introduced (Fig 3B). Multiplication ('·') of an excitatory synaptic interval $h^E$ by an inhibitory interval $h^I$ is defined as:

$$h^I \cdot h^E \overset{\text{def}}{=} h^E \setminus h^I \qquad (6)$$

The set minus operation "\" eliminates the points from the multiplicand $h^E$ that it has in common with the multiplier $h^I$ (Fig 3B*i* and 3B*ii*) thereby decreasing the multiplicand's length.

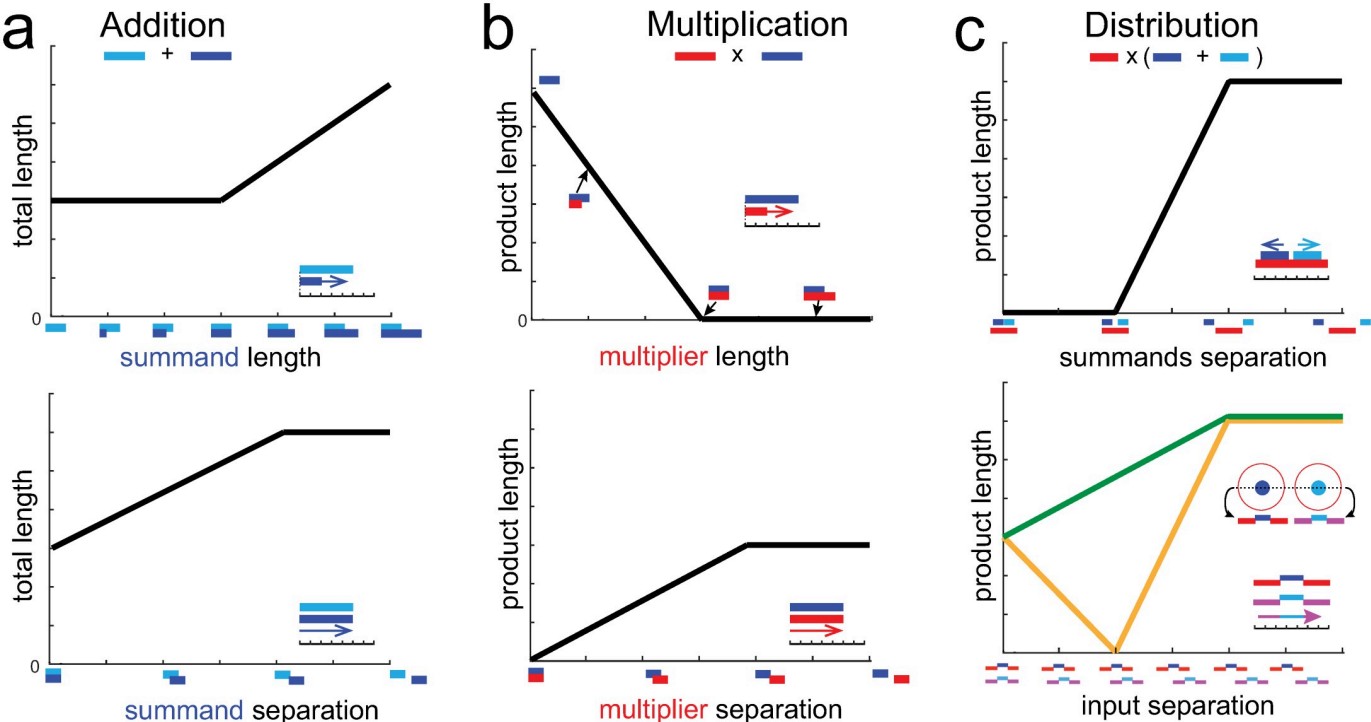

**Fig 4. Predicted effects of addition and multiplication.** *a*, *top*, addition of two intervals (cyan, blue) when the length of one interval (blue) is increased. *bottom*, one interval (blue) is shifted to the right of the other (cyan). *b*, top, multiplication of an excitatory interval (blue) by an inhibitory interval of increasing length (red). *bottom*, multiplication when the inhibitory interval is shifted to the right. *c, top*, effects of inhibition (red) on two excitatory intervals (blue, cyan). With the inhibitory interval at a fixed location, the distance between the excitatory intervals is increased systematically. *bottom*, Effects of two inputs configured as center-surround where the excitatory intervals (blue, cyan) are each flanked by two inhibitory intervals (red, magenta). One of the inputs is shifted systematically to the right of the other. Predicted product length when the inputs occur simultaneously (orange) and when calculated with inputs delivered sequentially.

Multiplication yields several results, depending on the relative locations and size of the multiplicand and the multiplier (S1 Appendix). If the excitatory (***E***) and inhibitory (***I***) intervals do not overlap, then the ***E*** interval is unaffected. If the intervals overlap ($h_\alpha^E$ and $h^I$), then the ***E*** interval is shortened ($h^I \cdot h_\alpha^E$ in *ii*). If the ***E*** interval ($h_\beta^E$) is completely within the ***I*** interval, the product is the empty set, indicating complete cancellation ($h^I \cdot h_\beta^E$ in *ii*). Multiplication can also change the starting points of the intervals and split an interval into two separate intervals. The algebraic properties of the multiplication operation are discussed in S1 Appendix.

If the excitatory and inhibitory inputs are co-activated (as in feedforward circuits), then the ***E*** and ***I*** intervals will be at the same location (same starting points) on the tonotopic axis but may have different lengths. Fig 4B (top panel) plots the predicted length of the product when the ***E*** interval is multiplied by ***I*** intervals of increasing lengths. The product length decreases and becomes zero when length of the ***I*** interval exceeds that of the ***E*** interval.

If the ***E*** and ***I*** inputs are independent of each other, the synaptic intervals could be at different locations on the tonotopic axis. Fig 4B (bottom panel) plots the length of the product when the starting point of the ***I*** interval (red) is shifted systematically to the right of the ***E*** interval (blue). The product length is zero when the ***E*** and ***I*** intervals overlap completely (separation = 0) and increases linearly as the overlap decreases, eventually plateauing to a constant value when the intervals become disjoint.

With addition and multiplication defined, the rules for combining the two operations can now be determined. A simple case is when a network receives two excitatory inputs that results

in synaptic intervals ($h_\alpha^E$, $h_\beta^E$) and a single inhibitory input that result in an inhibitory interval ($h^I$). This scenario would occur if binaural excitatory inputs that converge in a network are then acted on by local inhibitory neurons. When all three inputs are activated simultaneously, the intervals combine in neural space as $h^I \cdot (h_\alpha^E + h_\beta^E)$. It can be shown that multiplication is *left* distributive (S1 Appendix) so that:

$$h^I \cdot (h_\alpha^E + h_\beta^E) \quad = (h^I \cdot h_\alpha^E) + (h^I \cdot h_\beta^E) \tag{7}$$

Intuitively, this means that the effect of a single inhibitory input on two separate excitatory inputs can be calculated by computing the inhibitory effects on each separately and then adding the results. Multiplication, however, is not *right* distributive. Thus, given two inhibitory intervals ($h_\alpha^I$ and $h_\beta^I$) acting on a single excitatory interval ($h^E$):

$$(h_\alpha^I + h_\beta^I) \cdot h^E \neq (h_\alpha^I \cdot h^E) + (h_\beta^I \cdot h^E) \tag{8}$$

Fig 4C (top panel) plots the predicted length when two *E* intervals are multiplied by an *I* interval. The two *E* intervals (blue, cyan) are shifted, respectively, left- and rightward relative to the *I* interval (red). The product length is zero as long as the two *E* intervals are within the *I* interval. When the two *E* intervals reach and exceed the borders of the *I* interval, the product length increases and reaches a plateau when the *E* and *I* intervals become disjoint.

A common physiological scenario is when sound is composed of two pure tones and each tone results in an excitatory synaptic field surrounded by an inhibitory field (center surround inhibition, Fig 4C, bottom panel). The corresponding composite interval contains an excitatory interval that is flanked by two inhibitory intervals (inset). Letting the *I* -*E* -*I* interval triplet generated by each tone be described by $(h_\alpha^I + h_\beta^I) \cdot h_\epsilon^E$ and $(h_\gamma^I + h_\delta^I) \cdot h_\zeta^E$, the expression when both occur simultaneously is:

$$(h_\alpha^I + h_\beta^I + h_\gamma^I + h_\delta^I) \cdot (h_\epsilon^E + h_\zeta^E) \tag{9}$$

In Fig 4C (bottom panel, orange curve), the location of one composite interval is shifted rightward. When the composite intervals coincide (separation = 0), the product length is equal to that of a single excitatory interval. With increasing separation, the product length decreases towards zero but then increases, reaching a plateau when the excitatory and inhibitory components of each composite interval no longer overlap.

Because of the distributive properties, the effect of introducing two tones simultaneously cannot be predicted by introducing each separately and then combining the results. That is,

$$(h_\alpha^I + h_\beta^I + h_\gamma^I + h_\delta^I) \cdot (h_\epsilon^E + h_\zeta^E) \neq (h_\alpha^I + h_\beta^I) \cdot h_\epsilon^E + (h_\gamma^I + h_\delta^I) \cdot h_\zeta^E \tag{10}$$

The green curve in Fig 4C (bottom panel) is the predicted product length when the *I* -*E* -*I* triplet pairs are delivered separately and their product lengths subsequently summed. Intuitively, the curves differ because the effects of inhibition on the adjacent excitatory interval is absent; indeed, the result resembles that of adding two excitatory intervals (Fig 4A, bottom panel). A practical implication is that the intervals due to complex sound cannot be predicted by presenting individual tones separately (see Discussion).

## Simulations with spiking neurons

Key features of the mathematical model were examined with simulations performed on a 2 dimensional network model of spiking excitatory and inhibitory neurons in auditory cortex [32] (code available at https://github.com/AlexDReyes/ReyesPlosComp.git). This model was chosen because the firing properties of and connection schemes between *E* and *I*, which

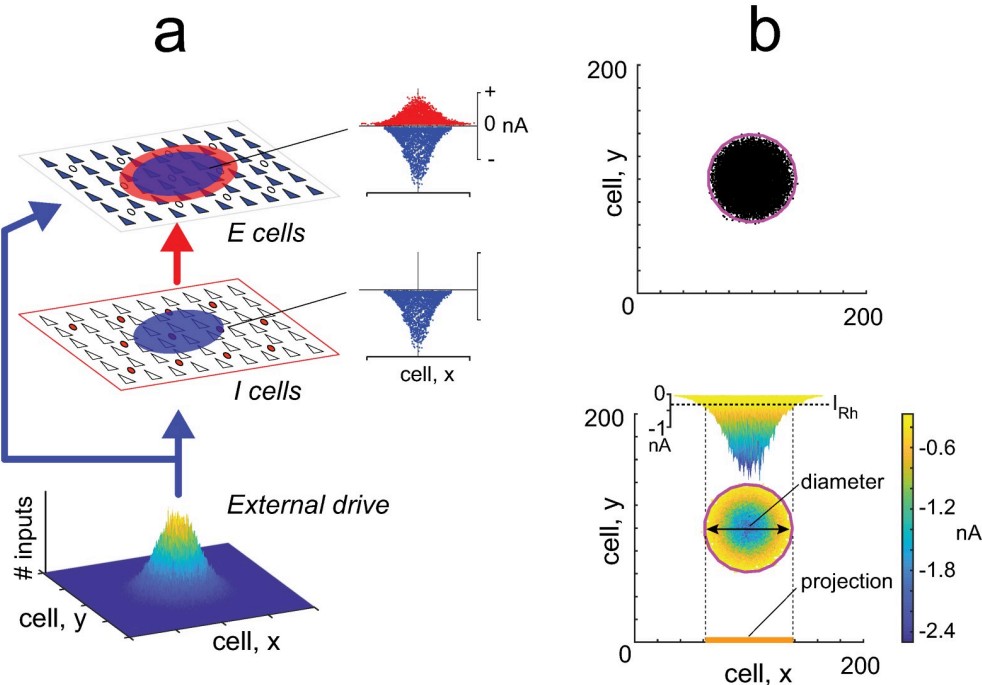

**Fig 5. Simulations with spiking neurons.** *a*, Network consists of excitatory and inhibitory neuron populations. An external drive evokes excitatory inputs in both populations (blue disks) and inhibitory inputs to the *E* cells (red disk). *insets*, profiles of excitatory (blue) and inhibitory (red) currents evoked in the *E* and *I* populations. *b, bottom*, synaptic field evoked in the network during a stimulus. The spatial extent of the synaptic field is quantified either by the diameter of a circle fitted to its outermost points (magenta) or by the length of its projection to the tonotopic axis (orange bar). *inset*, profile of net synaptic current generated in the *E* cell population. The perimeter of the synaptic field encompasses cells whose net synaptic current input exceeded rheobase ($I_{Rh}$). *top*, activated area contains cells that fired action potentials (dots).

determine the size of the synaptic fields, have been fully characterized experimentally [33] and can be modified readily. Extensive simulations also showed how the firing behavior is affected by the interaction of *E* and *I* cells [32]. Both *E* and *I* neuron population receive a Gaussian distributed excitatory drive from an external source (Fig 5A); the *E* cells in addition receive feed-forward inhibitory inputs from the *I* cells. Stimulation evokes Gaussian distributed excitatory inward currents in both populations and also inhibitory currents in the *E* cells (profiles of currents shown in insets). With brief stimuli, the recurrent connections between neurons [33] do not contribute significantly to activity in auditory cortical circuits [32] and were omitted. The region encompassing neurons that fire is henceforth referred to as the activated area (Fig 5B, top panel). The underlying synaptic field (bottom panel) is described by the area of the network where the *net* synaptic inputs to cells exceeded (were more negative than) rheobase, the minimum current needed to evoke an action potential ($I_{Rh}$, inset). As defined, the synaptic field is a composite of all the inputs, both excitatory and inhibitory, that are evoked during a stimulus. Both the activated area and the synaptic field are quantified either by the diameters of circles fitted to the boundary points (magenta) or by the length of their projections to one axis (orange bar). Note that the spatial dimensions have units of *cell number* (see Methods to convert to microns).

To test the addition operation, two external excitatory drives were delivered to the center of the network simultaneously (without inhibition). Increasing the width (by increasing the standard deviation $\sigma_\alpha$ of the external drive) of one stimulus, while keeping that ($\sigma_\beta$) of the other

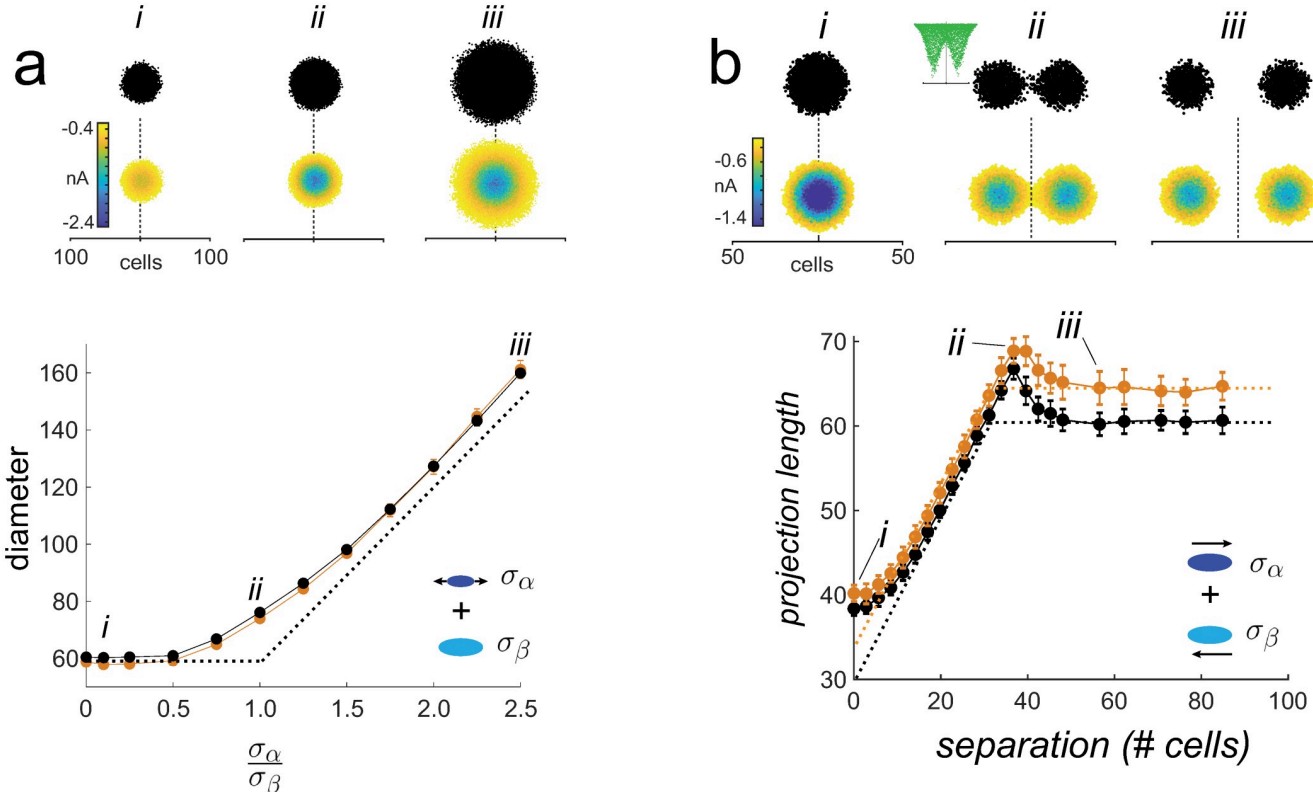

**Fig 6. Test of addition.** *a, top*, Activated area (top; 1 sweep) and underlying synaptic field (bottom; average of 25 sweeps). *i*, one stimulus. *ii-iii*, two stimuli delivered to the center of the network. The width of one input was systematically increased ($\sigma_\alpha$: 2- 45 cells) while that of the other ($\sigma_\beta$ = 20 cells) was kept constant. *bottom*, plot of synaptic field (orange) and activated area (black) diameters vs $\frac{\sigma_\alpha}{\sigma_\beta}$. Dashed curve is predicted relation. *b*, addition of two spatially separated excitatory inputs ($\sigma_\alpha = \sigma_\beta = 10$). *top, i–iii*, activated areas and synaptic fields with increasing stimulus separation. Inset in *ii* shows example of excitatory synaptic current profiles. *bottom*, projection length vs. separation distance for synaptic field (orange) and activated area (black). Dashed curves are predicted changes.

fixed, increased the diameters of the synaptic field and activated areas (Fig 6A, top panel, *i-iii*). As predicted, the diameters of the synaptic field (bottom panel, orange) and activated area (black) initially did not change but then increased as $\sigma_\alpha$ continued to widen. However, because the synaptic currents were Gaussian distributed (Fig 5A, bottom panel), the curve started to increase before $\sigma_\alpha$ became equal to $\sigma_\beta$ ($\frac{\sigma_\alpha}{\sigma_\beta} = 1$). When delivered simultaneously, the magnitude of the composite current increased, causing the region that exceeded rheobase to widen (Fig 6A, top panel, compare synaptic field evoked with a single stimulus (*i*) to that evoked with 2 stimuli (*ii*)). The diameter can be calculated from the standard deviations of the two inputs (diameter $\propto \sqrt{\sigma_\alpha^2 + \sigma_\beta^2}$).

To examine the addition of spatially disparate synaptic fields, two excitatory inputs were delivered at different distances from each other (Fig 6B, top panel). Consistent with the prediction, the projection lengths of the synaptic field (bottom panel, orange) and activated areas (black) increased with stimulus separation and reached a plateau when the two inputs became disjoint (*iii*). The projection lengths were greater than predicted (dashed lines) when the separation was small ($<$ 10 cells) and when the intervals were just becoming disjoint (at separation $\sim$40 cells, *ii*) due to the summation properties of the Gaussian distributed inputs discussed above.

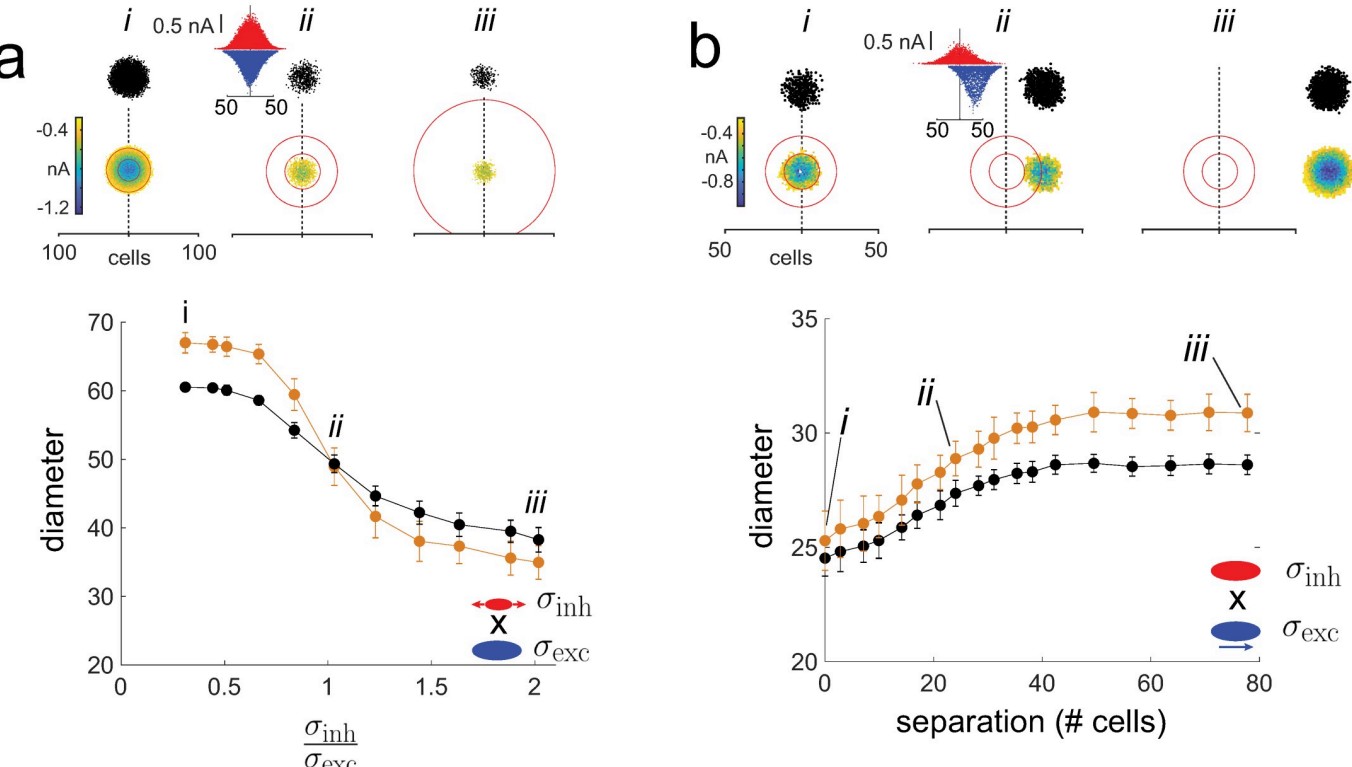

**Fig 7. Test of multiplication.** *a, top, i-iii* activated areas and synaptic fields evoked with excitatory ($\sigma_{exc}$ = 20) and inhibitory ($\sigma_{inh}$ = 2 − 45) inputs. The spatial extent of the inhibition is demarcated by the red circles (inner circle: 1 $\sigma_{inh}$; outer: 2 $\sigma_{inh}$). Inset in *ii* shows an example of excitatory (blue) and inhibitory (red) synaptic current profiles. *bottom*, plot of activated area (black) and synaptic field (orange) vs $\frac{\sigma_{inh}}{\sigma_{exc}}$. *b*, Same as in *a* except that the excitatory input ($\sigma_{exc}$ = 10) was shifted systematically to the right of inhibition ($\sigma_{inh}$ = 10). *bottom*, Diameters of activated area (black) and synaptic field (orange) plotted against separation between excitatory and inhibitory synaptic fields.

To test the multiplication operation, the **E** and **I** neurons were stimulated simultaneously, resulting in excitatory and inhibitory synaptic currents in the **E** cells (inset in top panel of Fig 7A*ii*). The width of the excitatory input ($\sigma_{exc}$) was kept constant while that of the inhibitory input ($\sigma_{inh}$) was increased systematically. As predicted, the diameter of the synaptic field (bottom panel, orange) and activated area (black) decreased with increasing $\sigma_{inh}$. However, the diameter asymptoted towards a non-zero value. Because the network was feedforward, the inhibitory input was delayed relative to excitation by about 10–15 ms; as a result, there was always a time window where excitation dominated [32]. The excitatory synaptic input was not canceled even when the inhibition was twice as wide (top panel, *iii*).

To examine multiplication of spatially disparate **E** and **I** inputs, the excitatory input was shifted systematically to the right of the inhibitory input (Fig 7B). As predicted, the diameters of the synaptic field (bottom panel, orange) and activated area (black) increased with the **E** -**I** separation and plateaued when the **E** and **I** inputs became disjoint (*iii*).

To examine how multiplication distributes over addition, two excitatory inputs and one inhibitory input were delivered simultaneously to the network (Fig 8A). This is the analog of the left hand side of (Eq 7). All three inputs were initially at the center and then with the inhibition stationary, the two excitatory inputs were shifted left and right (Fig 8A, top panel, *i-iii*). As predicted, the projection length of the synaptic field increased towards an asymptotic value (bottom panel, orange). To reproduce the right hand side of Eq 7, simulations were performed with inhibition, first with one of the excitatory inputs and then with the other; the resultant

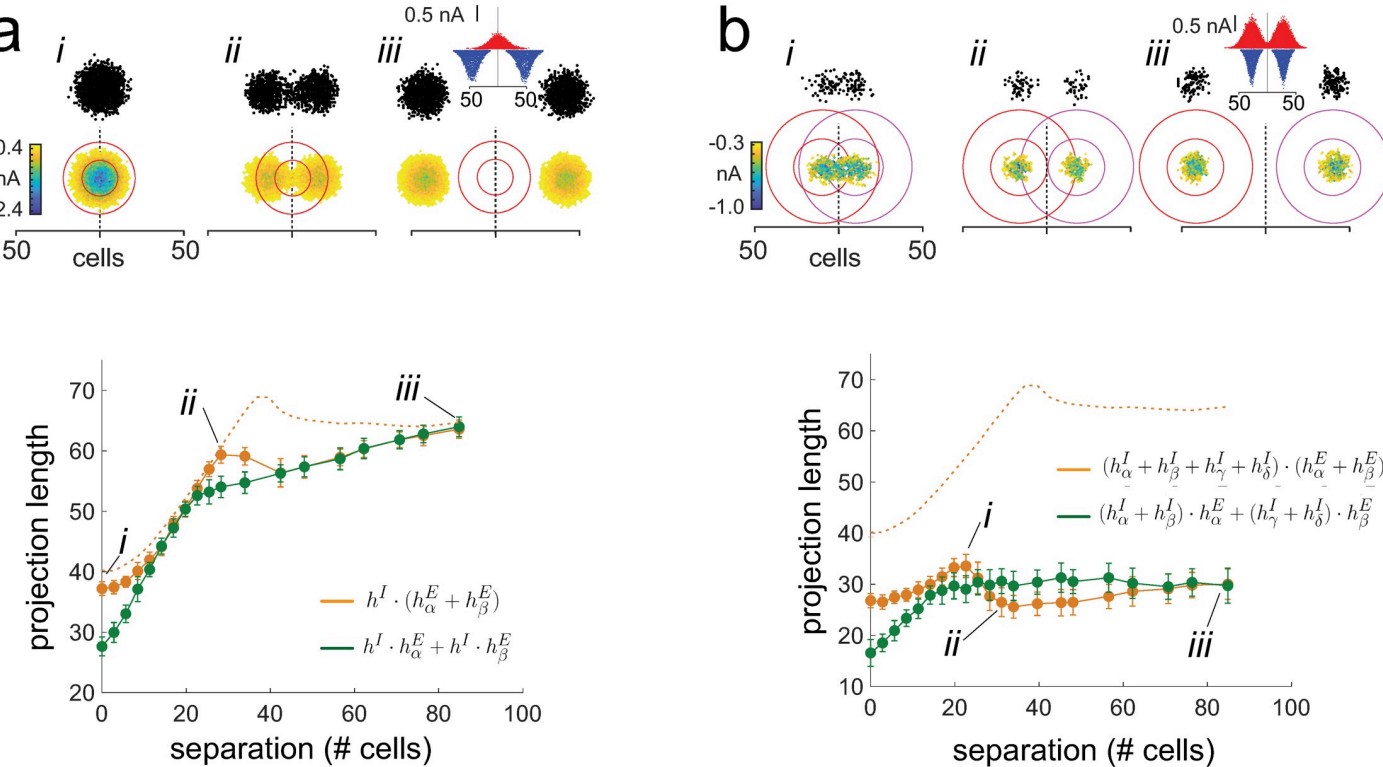

**Fig 8. Test of distributive properties.** *a, top, i-iii* Representative activated areas and synaptic fields generated by two excitatory inputs and a single inhibitory input ($\sigma_{exc} = \sigma_{inh} = 10$). With the location of inhibition fixed, the two excitatory inputs were separated systematically. The spatial extent of the inhibition is demarcated by the red circles (inner circle: 1 $\sigma_{inh}$; outer: 2 $\sigma_{inh}$). Inset in *iii* shows the excitatory (blue) and inhibitory (red) synaptic current profiles. *bottom*, plot of synaptic field projection lengths (orange) vs separation of the excitatory inputs. Green symbols are projection lengths obtained with the sequential stimulation protocol (see text). Dotted curve is with no inhibition. *b*, Simulations with two excitatory-inhibitory pairs, each with center-surround configuration (see inset in *iii*). *bottom*, legend as in *a* except that the green traces plot the projected lengths obtained when each input ($\sigma_{exc} = 10$, $\sigma_{inh} = 17$) was delivered sequentially (see text).

projection lengths of each were then summed (green). As was observed with simultaneous stimulation, the projection length increased with separation. The match was poor at small separations <10 cells (*i*) and at separation of ∼ 30 cells (*ii*) because the interaction between the Gaussian excitatory currents (see above) did not factor in when each input was delivered separately. The two curves were nearly identical at separations of 40–80 cells. In this range the ***E*** inputs were disjoint (as indicated by the plateauing of the excitation-only curve (dashed orange)) but still overlapped with the inhibitory synaptic field (the orange and green curves were below the excitation-only dotted curve).

Finally, the interaction of inputs with center-surround inhibition (Eq 9) was examined by delivering two excitatory inputs, each with associated inhibitory components (inset in Fig 8B, top panel, *iii*), to the network. The distance between the inputs was then increased systematically and the projection lengths measured (bottom panel, orange curve). At separations > 20 cells, the projection length of the synaptic field decreased to a minimum (*ii*) and then increased towards a plateau (*iii*), consistent with the prediction (Fig 4C, bottom panel). However, at separations < 20 cells, the length increased instead of decreasing; this is likely due to the interaction of the Gaussian distributed excitatory fields discussed above. To confirm that the same result cannot be obtained by presenting each stimulus separately (right hand side of Eq 10), each ***E*** -***I*** pair was delivered sequentially and the individual projection lengths summed. Unlike with simultaneous stimulation and consistent with the prediction (green curve in Fig 4C,

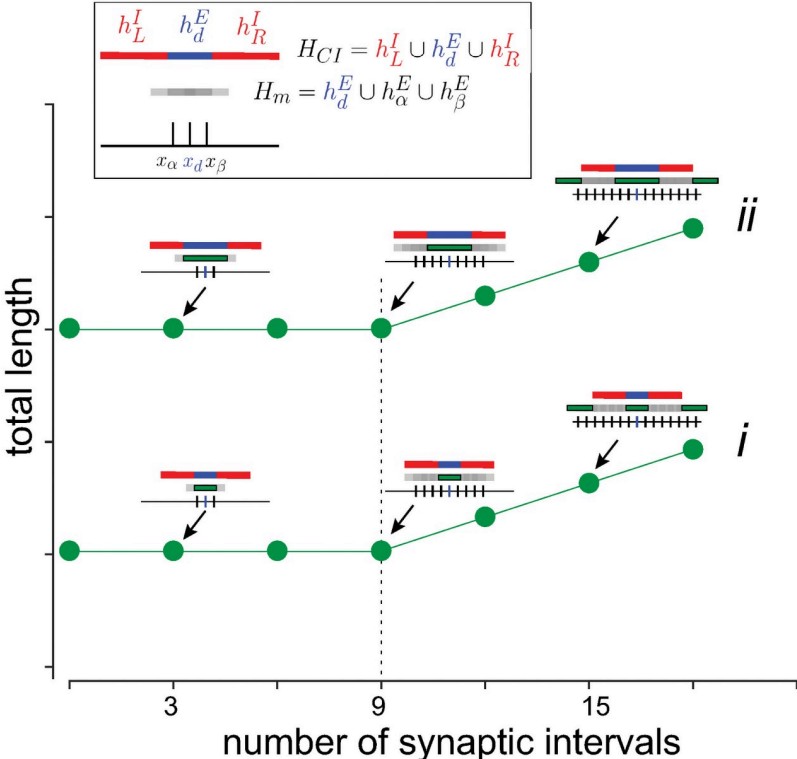

**Fig 9. Algebra of loudness summation.** Predicted interval lengths resulting from the interaction of multi-tone stimulus delivered simultaneously. *Boxed inset*, overlapping synaptic intervals ($H_m$, gray) generated by stimulus with 3 frequency components. Tic marks show location of interval centers ($x_\alpha$, $x_d$, $x_\beta$) along tonotopic axis. The dominant tone (blue) also generates two inhibitory side bands (red). Plot shows resultant length ($|h_l| = |(h_L^I + h_R^I) \cdot (H_m)|$) after the operations (see text) as the number of intervals in $H_m$ is increased (abscissa). Green bars in insets show portion of $H_m$ that was not cancelled by inhibition. Dotted vertical line marks deviation of curves from a constant value. Compare with Figs 9 of [37].

bottom panel), the projection length increased monotonically to a plateau without a dip (green curve).

## Application to loudness summation

In the auditory system, the perceived loudness of band limited noise or simultaneously presented tones depends on whether the frequency components are within the so-called critical band (CB) of frequencies [34–36]. An important property is that increasing the bandwidth of the noise does not increase the perceived loudness until the bandwidth exceeds CB, after which it increases linearly [37]. Moreover, this property is maintained at different sound intensities, indicating that CB does not change. The origin of the CB is unclear and there is debate as to whether it is peripheral involving mainly excitatory processes [38, 39], or central, which may also recruit inhibition [40–42]. The tonotopic axis is often divided into 24 CBs, each uniquely identified by the center frequency [35]. In the following, algebraic operations are used to describe features of loudness summation and to suggest network mechanisms.

A band-limited noise stimulus, or more generally a complex stimulus with multiple tones, may be expressed, after discretization, as a set of increasing frequencies, say: $F_m = \{f_1, f_2, \ldots, f_n\}$. The 'bandwidth' is defined as the difference between the highest and lowest frequency components ($BW = f_n - f_1$). In neural space, the stimulus results in an interval that is the union of individual

excitatory intervals generated by each tonal component: $H_m = \bigcup_{i=1}^{n}(h_i^E = [x_i, x_i + \lambda))$, where $\lambda$ is the length of each interval and is the same for all intervals.

The model assumes that for multi-tone stimulus, one of the tones is dominant and generates inhibitory intervals ($h_L^I$ and $h_R^I$) that abut an excitatory interval $h_d^E \in H_m$ with no overlap ($h_d^E \cap h_L^I \cap h_R^I = \emptyset$), as in a so-called lateral inhibitory configuration (see S1 Appendix for formal definitions). Physiologically, the dominant tone may correspond to the tone at the center of a CB [35] or to the tone with the lowest frequency, which has been shown to mask higher frequency components [43]. The union of these 3 intervals is defined to be the critical interval: $H_{CI} = h_d^E \cup h_L^I \cup h_R^I$. The boxed inset in Fig 9 shows the relationship between $H_m$ (gray), $h_d^E$ (blue), and the two inhibitory intervals (red). The length of the interval $h_l$ that results from the interaction of these intervals is given by $|h_l| = |(h_L^I + h_R^I) \cdot H_m|$ and is taken to be a proxy for loudness perception. As shown in S1 Appendix, $|h_l|$ is equal to $|h_d^E|$ as long as $H_m \subset H_{CI}$.

Fig 9 shows the result graphically when $H_m$ is lengthened by adding more tones to the stimulus. $|h_l|$ is constant ($= |h_d^E|$) until the number of components is such that $H_m$ exceeds the boundaries of the critical interval. In this example, the deviation occurs when the number of intervals, and hence the number of frequency components, exceed 9 (dotted vertical line). The CB is then $f_9 - f_1$.

Increasing the intensity of each component of $F_m$ causes an increase in the length of the interval components of $H_m$. As shown in S1 Appendix, the CB will not change provided that the lateral inhibitory configuration is maintained and the lengths of the inhibitory intervals are constant. Under this condition, $|H_m|$ and $H_{CI}|$ increase equally (compare lower and upper curves in Fig 9). Because $h_d^E$ increases, there is an increase in baseline (upward shift of curves) without a change in CB.

An all-excitatory version without inhibition will not reproduce the data: the critical interval would then be $h_d^E$ and since $h_d^E \in H_m$, $|h_l|$ will be greater than $|h_d^E|$ if $H_m$ has more than one component and will grow with increasing number of tonal components. Unlike the data, the curves would have no flat region.

The operations also describe a related experiment where instead of noise, the stimuli consisted of 4 tones whose frequency separations were varied systematically [37] (Fig B of S1 Appendix).

The above analysis elucidates the general requirements for loudness summation. While there is some evidence for a dominant tone [43] and inhibitory processes [42], the extent of the inhibitory intervals is less clear and is likely to reflect the combined effect of the individual excitatory and inhibitory intervals generated by other tones in the stimulus. The precise mechanisms needs to be systematically explored with more detailed analyses, simulations, and experiments.

## Discussion

The aim of this study was to develop a mathematical framework for a place-code and derive the underlying principles for how tones of varying frequencies and intensities are represented, assembled, and modulated in networks of excitatory and inhibitory neurons. The analyses are not intended to replicate the detailed aspects of biological networks and dynamic behavior but rather to clarify the minimal conditions that must be met for a viable place coding scheme, to aid in the interpretation of experimental data, and to provide a blueprint for developing computational models. The advantage of this formal approach is that it ensures that the terms and advantages/limitations of a purely place-coding model are defined precisely, providing a foundation for examining the role of other auditory cues that enhance coding and perception

(see below). In addition, the mathematical rules effectively constrain the computations that may be performed with a purely place code.

## Place code framework in auditory processing: Evidence and implications

The model has several implications with regards to auditory processing. In this section, the advantages of the place coding framework are discussed and experimental data are interpreted within the context of the mathematical framework.

**Representation of frequency and sound pressure.** A key feature of the model is that the 'functional unit' of neural space is a set of contiguous neurons that have flexible borders. The associated mathematical architecture is a collection of half-open intervals of varying lengths. The model provides a framework for encoding both frequency and intensity (or sound pressure) with a purely place-coding scheme. This is advantageous for brief stimuli where firing rate and spike timing [8, 12, 19] may not be available (see Introduction). Some information may be carried by single spike latency [20]; however, spike latencies depend on other variables besides frequency and does not appear to have the dynamic range to represent the full range of audible sound pressure levels [44]. Frequency and intensity discrimination does improve with stimulus duration, suggesting that the other variables play complementary roles in improving coding and perception [9–12, 22].

A network with flexible functional units is also advantageous for maintaining both high resolution frequency and pressure representations. This can be appreciated by comparing the resolutions attainable with the classical columnar organization [23, 25] (the stimulus is assumed brief so that firing rate information is unavailable; see Introduction). In this scheme, the neural space is divided into non-overlapping columns with fixed dimensions and distinct borders. The frequency of a stimulus is encoded by the location of the active column and sound pressure by the number of active neurons within the column (i.e. population rate code). The relation between the maximum number of achievable frequency and sound pressure levels is given by $|\mathcal{P}| = \frac{N_h}{|\mathcal{F}|}$ (see S1 Appendix). Intuitively, to maximize the number of frequency levels, the columns should be as small as possible so that more can fit along the tonotopic axis; however, this reduces the number of pressure levels that can be encoded because there are fewer neurons within a column. In contrast, for a network with flexible borders, the relation is: $|\mathcal{F}| = N_h - |\mathcal{P}| + 1$. Fewer neurons ($N_h = |\mathcal{F}| + |\mathcal{P}| - 1$) are needed to represent the full range of frequency and pressure levels as compared to columns ($N_h = |\mathcal{F}| \times |\mathcal{P}|$).

The advantage of a columnar organization is that the components of a multi-frequency stimulus remain separated in neural space. With flexible units, two intervals generated by two tones with small frequency differences and/or high intensities can fuse into a single interval and hence be perceived as a single tone. As discussed below, ambiguities in perception of complex stimulus are more consistent with a flexible unit organization.

**Relation between $\Delta f$ and frequency difference limen.** In the model, the acoustic space is discretized to reflect the resolution limits on frequency and intensity perception imposed by neural space composed of neurons. The number of frequency levels and $\Delta f$ is determined by the number of intervals that can be contained within the neural space (Eq 2). Though the model was introduced with $\Delta x$ equivalent to the diameter of a cell in a single layer (Fig 2), $\Delta x$ (and hence $\Delta f$) can be much smaller if several layers of neurons are considered (Fig A of S1 Appendix).

The frequency difference limen ($\Delta f_{DL}$), gives the smallest difference in frequency of two tones that can be discriminated by subjects. The measured $\Delta f_{DL}$ does not have a fixed value but depends on a number of stimulus parameters including duration, intensity, and test frequencies [10, 45]. Moreover, $\Delta f_{DL}$, which is related to the psychophysical measure of sensitivity

('d-prime', [46, 47]), is affected by unspecified sources of internal noise within subjects such as trial-to-trial variability in pitch perception [48]. For these reasons, $\Delta f_{DL}$ is likely to be larger than $\Delta f$. Thus, $\Delta f$ may be viewed as the lower bound for $\Delta f_{DL}$ for a purely place-coding scheme that would be realized under optimal, noiseless conditions.

**Addition operation.** The addition operation applied to synaptic intervals is defined as their union: $h_\alpha + h_\beta \overset{\text{def}}{=} h_\alpha \cup h_\beta$. An important consequence is that if the intervals overlap, they will fuse into a single, longer interval. Under physiological conditions, this would occur if tones of a multifrequency stimulus have small differences in frequencies. This is in line with psychophysical experiments, which show that subjects perceive tones with small differences in frequencies as a single tone [43, 49] and have difficulties distinguishing the individual components of a multi-frequency stimulus [50, 51].

Another consequence is that addition of two overlapping non-empty intervals is sublinear: $|h_\alpha| + |h_\beta| > |h_\alpha + h_\beta|$. If one interval is also a subset of the other ($h_\alpha \subset h_\beta$), then the sum is equal to the larger of the two intervals: $|h_\alpha| + |h_\beta| = |h_\beta|$. This scenario would occur when binaural inputs converge onto a common site. Consistent with the prediction, electrophysiological recordings from neurons in inferior colliculus show that the frequency response areas (FRAs, assumed to be representative of activity spread, see below) evoked binaurally is equal to the larger of two responses evoked monaurally [52]. Similarly, assuming that loudness perception is linked to the length of the interval, a possible psychophysical analog is that a tone presented binaurally to a subject is perceived to be less than twice as loud as monaural stimulation [53]. The apparent sublinear effects can be explained by the properties of addition operation, though inhibitory processes may also contribute.

**Multiplication operation and distributive properties.** Multiplication of two synaptic intervals is defined as the set minus operation: $h_\alpha \cdot h_\beta = h_\beta \setminus h_\alpha$. The operation removes from the multiplicand ($h_\beta$) elements that it has in common with the multiplier ($h_\alpha$), thereby shortening it. The effect of inhibition can be inferred from the FRAs of neurons. Applying GABA blockers causes the FRAs to widen [54, 55]. If the FRA can be used as a proxy for the spatial extent of activated neurons (see below), then the result is consistent with inhibition shortening the synaptic intervals.

The manner in which multiplication distributes over addition has important implications for combining information from multiple sources. In auditory cortex, excitatory pyramidal neurons receive convergent afferent inputs from the thalamus and other pyramidal cells [56, 57]. The two afferents also appear to innervate a common set of local inhibitory neurons [33, 57]. The fact that multiplication is left distributive (Eq 7) means that the effect can be estimated by measuring the effects of inhibition ($h^I$) on each excitatory inputs ($h^E_{ctx}$, $h^E_{thal}$) separately and then summing the results: $|h^I \cdot (h^E_{ctx} + h^E_{thal})| = |h^I \cdot h^E_{ctx}| + |h^I \cdot h^E_{thal}|$. However, because multiplication is not right distributive (Eq 8) a similar approach cannot be used to examine two sources of inhibition acting on a single excitatory interval. The analyses, for example, suggest that the combined effects of two types of inhibitory neurons on excitatory cells [31] should be examined by activating both interneurons simultaneously rather than separately.

More generally, the representation of complex sound with a place coding scheme cannot be predicted by combining the representations of individual components if the inhibition generated by each component interact. As shown in Eq 10 and Fig 4C (bottom), the response of two tones presented simultaneously is not a simple combination of the responses to each tone separately. It should be emphasized that this conclusion was derived mathematically from the distributive properties; it is not trivially related to non-linearities contributed by e.g. inhibitory conductances or voltage gated channels since the model has no biophysical variables.

## Assumptions and limitations

As evidenced by cochlear implants, at least rudimentary pitch perception can be achieved with a purely place code [6, 7]. However, extracting the auditory features completely requires additional cues. Firing rate and spike timing information has been shown to enhance coding and perception [8, 12, 19–22]. Indeed, some neurons are specialized for extracting precise temporal information [16, 58]. Moreover, frequency and intensity discrimination improves with stimulus duration [9–12], indicating the contribution of dynamic processes at the synaptic [59] and network [32] levels. Sound localization [60] and beat generation [5], both of which use phase information, cannot be implemented with a purely place code. Perception of a fundamental frequency absent from a harmonic (missing fundamental [61]) also cannot be explained with a place code as the model predicts that only intervals generated by sound can be perceived. Finally, variables that affect the intervals and operations on intervals such as non-linearities due to biophysical properties of cells (Figs 6 and 7, see below) and cochlea [62] are absent from the model. The formal approach used here can in principle be used to incorporate these variables, with the place-coding framework as a starting point.

The mathematical model is based on two salient features of the auditory system. One is that the neural space is organized tonotopically. Tonotopy has been described in most neural structures in the auditory pathway, from the cochlea and auditory nerve [2, 3, 63, 64] to brainstem areas [4, 65, 66] to at least layer 4 of primary auditory cortex. Whether tonotopy is maintained throughout cortical layers is controversial, with some studies (all in mice) showing clear tonotopy [67–70] and others showing a more 'salt-and-pepper' organization [70–72]. A salt-and-pepper organization suggests that the incoming afferents are distributed widely in the neural space rather than confined to a small area. The model needs a relatively prominent tonotopy to satisfy the requirement that synaptic intervals encompass a contiguous set of cells.

A second requirement is that the size of the synaptic interval and activated area increase with the intensity of the sound. Intensity-related expansion of response areas occurs in the cochlea [27, 28, 73] and can also be inferred from the excitatory frequency-response areas (FRAs) of individual neurons. The excitatory FRAs, which document the firing of cells to tones of varying frequencies and intensities, are typically "V-shaped". At low intensities, neurons fire only when the tone frequencies are near its preferred frequency (tip of the V). At higher intensities, the range of frequencies that evoke firing increases substantially [68, 74]. If adjacent neurons have comparably-shaped FRAs but have slightly different preferred frequencies, an increase in intensity would translate to an increase in the spatial extent of activated neurons.

For mathematical convenience, the location of the synaptic intervals was identified by the leftmost point (closed end) of the interval, with increases in intensity signaled by a lengthening of the interval in the rightward (high frequency) direction. Similar behavior has been observed in the cochlea albeit in the opposite direction: an increase in the intensity causes response area to increase towards low frequency region of the basilar membrane while the high frequency cutoff remains fixed [3, 28, 73]. The choice of the leftmost point to tag the interval is arbitrary and any point in the interval will suffice provided an analogous point can be identified uniquely in each interval in the set. Experimentally, using the center of mass of active neurons as the identifier might be more practical.

For simplicity, both $\Delta f$ and $\Delta p$ are kept constant along the tonotopic axis, which is inaccurate because the range of frequencies and sound pressure changes with frequency and sound pressure level. To represent the full ranges, the frequency and pressure can be transformed into an octave and decibel scales prior to mapping to neural space.

The algebraic operations were derived from set theoretic operations and the magnitude of the underlying synaptic inputs were irrelevant. Under biological conditions, the input magnitude determines the degree to which biophysical, synaptic, and network processes become engaged, which will affect the length of the synaptic intervals and activated areas. Not surprisingly, the results of the network simulations deviated quantitatively from the mathematical predictions in some regimes (compare Fig 4 to Figs 6 and 7). Most of the discrepancies in the simulations were because the magnitudes of the synaptic inputs were Gaussian distributed along the tonotopic axis. In biological networks, the discrepancies may be exacerbated by the presence of threshold processes such as regenerative events [75, 76]. The underlying algebraic operations may be obscured in regimes such as these.

The model incorrectly assumes that the strength of inhibition is sufficiently strong to fully cancel excitation. This facilitated analysis because the effect of multiplication depends solely on the overlap between the multiplicand and multiplier. As the simulations with the feedforward network showed, the excitation cannot be fully canceled by inhibition owing to synaptic delay. Moreover, the balance may be spatially non-homogeneous: in center-surround suppression, excitation dominates at the preferred frequency with inhibition becoming more prominent at non-preferred frequencies [54, 55, 74]. To apply multiplication to biological systems, it may be necessary to define empirically an "effective" inhibitory field that takes into account for $E$-$I$ imbalances.

For convenience, the simulations that were used to test the analyses predictions used a network model based on cortical circuits where the properties of the cells and patterns of connections betwen $E$ and $I$ cells have been fully characterized [32, 33]. However, the results should generalize to other network types provided the stimuli are brief (50 ms) so that cells fire only a single action potential. The mathematical model treats neurons as binary units and so only the first action potential is important. Hence, if the stimulus is brief and suprathreshold, the results obtained with a network consisting of e.g. repetitively firing cortical neurons [15, 33] or transiently firing bushy cells [58] will be qualitatively similar. The results are likely to differ with longer duration stimuli, which would allow various time- and voltage-dependent channels to become active and engage recurrent connections. It would also be important to confirm the operations for combining tones using cochlear/auditory nerve models that implements tonotopy derived directly from the basilar membrane [77, 78].

## Methods

Simulations were performed with a modified version of a network model used previously [32]. Briefly, the model is a 200 x 200 cell network composed of 75% excitatory ($E$) and 25% inhibitory ($I$) neurons. The connection architecture, synaptic amplitudes/dynamics, and intrinsic properties of neurons were based on experimental data obtained from paired whole-cell recordings of excitatory pyramidal neurons and inhibitory fast-spiking and low threshold spiking interneurons [33]. For this study, the low-threshold spiking interneurons and the recurrent connections between the different cell types were removed, leaving only the inhibitory connections from fast spiking interneurons to pyramidal neurons. The connection probability between the inhibitory fast-spiking cells and the excitatory pyramidal cells was Gaussian distributed with a standard deviation of 75 $\mu$m and peak of 0.4 [33].

Both $E$ and $I$ cells received excitatory synaptic barrages from an external source. The synaptic barrages to each cell (50 ms duration) represented the activity of a specified number of presynaptic neurons. The average number ($n_{in}(x, y)$) of inputs that each neuron at location $x, y$ received followed a Gaussian curve so that cells at the center of the network received more inputs (Fig 5A, bottom). For each run, the number was randomized by drawing a number

from a Gaussian distribution with mean $n_{in}(x, y)$ and a standard deviation $0.25 * n_{in}(x, y)$ so that the synaptic fields and activated areas varied from trial to trial. Excitatory synaptic currents were evoked in the **E** and **I** cell populations and inhibitory synaptic currents in the **E** cell population after the **I** cells fired (insets in Fig 5A). The spatial extents of the synaptic inputs were varied by changing the standard deviations of the external drive. In some simulations, the **E** and **I** cell populations were uncoupled and received separate inputs that could be varied independently of each other. The neurons are adaptive exponential integrate-and-fire units with parameters adjusted to replicate pyramidal and fast spiking inhibitory neuron firing (see [32] for the parameter values).

The synaptic field was defined as the area of the network where the net synaptic currents to the cells exceeded rheobase, the minimum current needed to evoke an action potential in the **E** cells ($I_{Rh}$, inset in Fig 5B, bottom panel). $I_{Rh}$ was estimated by calculating the net synaptic current near firing threshold ($V_\theta$): $I_{net} = g_{exc} * (V_\theta - E_{exc}) + g_{inh} * (V_\theta - E_{inh})$ where $g_{exc}$, $g_{inh}$ are the excitatory and inhibitory conductances, respectively, and $E_{exc} = 0$ mV, $E_{inh} = -80$ mV are the reversal potentials. For the **E** cells, rheobase is approximately -0.27 nA.

The spatial extent of the synaptic field or activated area was quantified as the diameter of a circle fitted to the outermost points (maroon circles in Fig 5B). In simulations with multiple components, the spatial extents were quantified as the total length of the projection onto the tonotopic axis (orange bar in Fig 5B, bottom panel). The diameters and lengths have units of cell number but can be converted to microns by multiplying by 7.5 $\mu$m, the distance between **E** cells in the network. For all plots, the data points are plotted as mean +/- standard deviation compiled from 20–100 sweeps.

## Supporting information

**S1 Appendix. Detailed description of mathematical analyses and proofs.** Fig. A: Projections of multiple layers of staggered neurons on tonotopic axis decreases $\Delta x$. Fig. B: Algebra of loudness summation applied to stimuli consisting of 4 tones with equally spaced frequencies. (PDF)

## Acknowledgments

I thank L-S Young for her insightful critiques and A. Bose for commenting on an early version of manuscript.

## Author Contributions

**Conceptualization:** Alex D. Reyes.

**Data curation:** Alex D. Reyes.

**Formal analysis:** Alex D. Reyes.

**Funding acquisition:** Alex D. Reyes.

**Investigation:** Alex D. Reyes.

**Methodology:** Alex D. Reyes.

**Project administration:** Alex D. Reyes.

**Resources:** Alex D. Reyes.

**Software:** Alex D. Reyes.

**Supervision:** Alex D. Reyes.

**Validation:** Alex D. Reyes.

**Visualization:** Alex D. Reyes.

**Writing – original draft:** Alex D. Reyes.

**Writing – review & editing:** Alex D. Reyes.

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
