## [Decision Letter · Decision Letter 0]

1 Oct 2020

Dear Dr. Reyes,

Thank you very much for submitting your manuscript "Mathematical framework for place coding in the auditory system" (PCOMPBIOL-D-20-01254) for consideration at PLOS Computational Biology. As with all papers peer reviewed by the journal, your manuscript was reviewed by members of the editorial board and by several independent peer reviewers. Based on the reports, we regret to inform you that we will not be pursuing this manuscript for publication at PLOS Computational Biology.

The reviews are attached below this email, and we hope you will find them helpful if you decide to revise the manuscript for submission elsewhere. We are sorry that we cannot be more positive on this occasion. We very much appreciate your wish to present your work in one of PLOS's Open Access publications. 

Thank you for your support, and we hope that you will consider PLOS Computational Biology for other submissions in the future.

Sincerely,

Lyle J. Graham

Deputy Editor

PLOS Computational Biology

Reviewer's Responses to Questions

**Comments to the Authors: **

Reviewer #1: The author presents a mathematical framework to address the place coding of frequency and intensity in the auditory system. The analysis focusses on the coding of frequency and intensity for brief stimuli where neural responses can be viewed as binary. The author defines the acoustic and neural spaces mathematically and considers conditions that support a place code. The manuscript is clearly written. However, I have a few questions about the model formulation.

General comments

The work aims to illustrate the conditions that support a place code for frequency and intensity. However, the results do not show the code performing in a way that matches or predicts psychophysical measurements. In describing equation 4 (line 120) it is stated that the mapping can be made to be bijective by adjusting the number of elements in each space. This can certainly be done if there are no constraints. Is it possible to start from psychophysical measurements and predict something about the neural representation, or vice versa?

A fundamental aspect of the analysis is the partitioning of the frequency and pressure spaces into a discrete set of equivalence classes. However, I have trouble understanding the definition of the equivalence classes in relation to the notion of perceptual discriminability. For example, in the equivalence classes f_min and f_min + df - epsilon are equivalent for any 0 < epsilon <= df, but f_min + df – epsilon and f_min+df are different for any epsilon > 0. This says that the JND is essentially 0 for frequencies at the boundary of the intervals. This would also suggest that JND measurements should show a strong asymmetry for test frequencies above or below a reference frequency. Please clarify how the defined equivalence classes relate to perceptual discriminability.

The nature of noise in the model is unclear. The input to the network is described as Gaussian distributed. Are the inputs drawn from a Gaussian distribution, or does the strength of the input follow the shape of a Gaussian curve? Please discuss the nature of noise in the model and how the model may be limited if noise is not considered. 

The simulation with spiking neurons tests the predictions of the addition and multiplication operations on an adaptive exponential integrate and fire model with parameters that are based on pyramidal neuron responses. Given the diversity of response types in the auditory system, the simulation does not appear to be a sufficient test of the model as a general place coding model that would apply throughout the auditory system. Is it intended that the model applies specifically to a population of neurons with the parameters used in the simulation? It would strengthen the results to show that the spiking simulation is robust to changes in neuron parameters.

Specific comments

Line 92: Please discuss why dx is constant. Shouldn’t dx depend on x if different frequency ranges are differentially represented in the neural space?

Lines 107-109: It is stated that frequency/pressure takes on an uncountable number of values, but the audible range is bounded. A bounded interval is still uncountable. Did you mean unbounded instead of uncountable, or is the discretization of frequency/pressure what is being referenced here?

Line 113: Please discuss the motivation for df and dp being independent of f and p, respectively, when the just noticeable difference frequency and pressure depend on the reference frequency and pressure. 

Lines 248-251: It is stated that the prediction was for no change when sigma_a was smaller than sigma_b, but in parentheses this is written as (sigma_a/sigma_b < 0.5). Should the condition (sigma_a/sigma_b < x) be equivalent to the condition that is written?

Supplementary material equation 1, equation 4a: It appears that the intervals h_f and h_p have f and p as arguments, respectively. The author might consider different notation for the frequency and pressure intervals because it is not clear whether h_2 is a frequency or pressure interval, for example. There is a similar ambiguity for H_f and H_p. It is typically clear from the context what is being referred to, but the author might consider revising the notation.

Supplementary material equation 18 bottom: I believe that a_f should be a_f_alpha or a_f_beta.

Reviewer #2: I do not believe that the modelling framework of this paper is adequate to draw the stated conclusions.

Firstly, there appear to be some fundamental problems in terms of the definition of the acoustic space. As far as I understand, the acoustic space is defined as a discrete 2d grid where a given pure tone of frequency f and amplitude p is mapped to a point in this grid. A pure tone is then represented essentially by a matrix where all elements are 0 except a single element which is 1 (I state it like this because it seems much simpler to get your head round that than the version in the paper, and entirely equivalent). If we were only talking about a single pure tone, then the stimulus space would be the set of all matrices with this property. However, it seems that combinations of tones are allowed, and an addition operation is defined on the acoustic space in the supplementary material (eqs 17-18), so that simultaneously playing two tones at different frequencies and amplitudes gives you a matrix with two 1s. However, there is a major problem here that can be illustrated with the very simple example of adding two tones at the same frequency.

Two tones with the same frequency f and amplitude p added together will give a new tone with the same frequency f and depending on their phase relationship any amplitude between 0 (out of phase) to 2p (in phase). Equation 18 only allows for an output amplitude of p. I am not sure therefore what the addition operation in the acoustic space is supposed to represent, but it isn't sounds played simultaneously. This alone seems to fatally undermine the rest of the paper. If something as basic as adding two tones together isn't handled correctly, we would need a very strong argument and clear reasoning that this framework has anything to say about real auditory perception, and there is no such argument in the paper.

The discussion (L311-319) states that three key findings, but none of them are findings, they are rather the definitions of the model. There is no 'finding' that neural space is a set of contiguous neurons, it's the definition of the model and it is not an accurate model. There is no 'finding' that operations in neural space must be union and set difference, it's another definition of the model.

I'm generally very sympathetic to what I see as the aim of this paper, which is to make a mathematical model which - even though it oversimplifies - is able to throw light on some fundamental aspect of the problem. However, I don't see any of that here. I see a framework which is not only oversimplified but not even clearly defined, and "conclusions" drawn which are just restatements of the incorrect assumptions of the model.

Reviewer #3: In the manuscript by A. Reyes, the author deals with the challenge imposed by the observation that animals can decode both the frequency and the amplitude of sound within a short period of time after exposure. This observation is seemingly incompatible with rate and temporal codes. The author considers a different (previously proposed) model of a “place code” in which the frequency response is defined by tonotopy within a network and amplitude response is defined by the extent of activation within the same network. The author proposes an algebraic formalist to describe the mapping between the sound and the neural space. Borrowing from set theory, the author introduces operations of addition and multiplication on this place code. Finally, the author compares results of these operations with network simulations.

Major critique

It is somewhat unclear what the main findings are, and there is substantial confusion between what the author calls “findings” of the study and what are actually just the definitions and assumptions of the model that the author proposes.

Key finding are outlined in lines 311-319 of the discussion. There are three. The first finding about the “function unit” with flexible borders (lines 311-314) is not truly a finding, but a definition of the mathematical framework proposed by the author. The second finding about the range and resolution dictated by the network architecture (lines 314-316) is a trivial consequence of using a place code: It is a given that network architecture in which different neurons respond to different frequencies will dictate the limitations of the code. In fact, the author outlines the intuition behind this conclusion in the Introduction. Finally, the operations of addition and multiplication (lines 316-317) are again not a finding, but part of the framework proposed by the author. It is not clear whether these operations are simply a formalism to describe the abilities and limitations of the place code, or whether these algebraic operations provide a new insight into these abilities and limitations.

Perhaps the manuscript is not intended to outline findings about neural networks, but is rather intended to introduce a formalism that may be useful for describing place coding. In this case, it is unclear in what way this formalism is useful. Does it allow describing phenomena that could not be described before? Does it reveal insights or make predictions about auditory circuits?

The author appears to make two predictions. One is that the place code is “sufficient for representing sounds” (line 329). It is unclear how this conclusion is derived, and what “sufficient” even means. Clearly, the manuscript itself states that there are limitations of the place code with regard to the resolution of sounds and ambiguity of complex sounds (lines 139-143).

A second prediction is that the response to a complex sound is not simply a combination of responses to the simple sounds that it consists of. In other words, the manuscript states that responses are nonlinear. This appears to be a trivial consequence of using a nonlinear process of inhibition, as defined by the manuscript itself.

In summary, I was left with major confusion about the main premise of the manuscript, the main results, and the predictions.

Minor comments

- The assumption that rate coding does not work on very short timescales is questionable. Even if an individual neuron is binary (spike or no spike) within a short time interval, the firing rate can be measured in a large enough population of neurons as the probability of spiking – i.e., the fraction of neurons with a certain preferred frequency that responded within a short time window.

- It is unclear why the manuscript defines frequency as the leftmost endpoint of an interval. Shouldn’t it be the center of an interval? I.e., louder sounds should activate frequencies both to the left and to the right of a preferred frequency.

- Line 41: “unlikely” should actually be “likely”

- Line 137: < (less than) should be less than or equal to

- Line 215: the use of subscripts is confusing because two variables with the same subscript actually have no relationship to one another (in particular h_beta^I and h_beta^E)

- Fig.6 has confusing usage of diameter and projection length, which both appear to be related to the extent of activation.

**Have all data underlying the figures and results presented in the manuscript been provided?**

Reviewer #1: Yes

Reviewer #2: No: There is no code or data given. There's no experimental data reported, so that doesn't need to be provided, but I'm not sure if code needs to be made available or not.

Reviewer #3: Yes

PLOS authors have the option to publish the peer review history of their article (what does this mean?). If published, this will include your full peer review and any attached files.

Reviewer #1: No

Reviewer #2: No

Reviewer #3: No

---

## [Decision Letter · Decision Letter 1]

3 May 2021

Dear Dr. Reyes (Alex),

Thanks for your patience. I was obligated to recruit a new reviewer for this round, but I think that she/he took into account the process from the beginning.

It's clear that both reviewers like what you are trying to do. At the same time I think that their comments pose a challenge, which is why this is a Major Revision, and I leave it to you to decide if you would like to submit a revision.  The rest is the form letter.

Best,

Lyle

Thank you very much for submitting your manuscript "Mathematical framework for place coding in the auditory system" for consideration at PLOS Computational Biology.

As with all papers reviewed by the journal, your manuscript was reviewed by members of the editorial board and by several independent reviewers. In light of the reviews (below this email), we would like to invite the resubmission of a significantly-revised version that takes into account the reviewers' comments.

We cannot make any decision about publication until we have seen the revised manuscript and your response to the reviewers' comments. Your revised manuscript is also likely to be sent to reviewers for further evaluation.

Sincerely,

Lyle J. Graham

Deputy Editor

PLOS Computational Biology

Lyle Graham

Deputy Editor

PLOS Computational Biology

Reviewer's Responses to Questions

**Comments to the Authors:**

Reviewer #1: I appreciate the goal of developing a mathematical approach to studying place coding in the auditory system. The author improved the manuscript with the revision, but there remain points that should be clarified.

Comments

1. Although my comment 3 from the first review was addressed in the manuscript, the issue continues to be a point of confusion. The author now states that experiments should be performed to test the hypothesis that the JND should range from being infinitesimally small to being equal to ∆f. While it would be helpful to directly test the hypothesis, it seems unlikely that such a striking phenomenon would not have been noted in the many existing studies of frequency JND. Is it possible to revise the framework so that the JND would not be sensitive to where the test frequencies fall within the intervals?

2. The mathematical framework should make statements about coding as precise as possible. With this view, the notion that df and dp are loosely related to the JND is unsatisfying. Can the nature of this relationship be made explicit?

3. Please clarify the definition of the neural and acoustic spaces. It appears that the spaces are defined as the collection of half-open intervals. If this is the case, then the space is not closed under the addition operation (union) because the union of two disjoint half-open intervals is not a half-open interval. It is not clear whether the spaces are the collection of half-open intervals or the collection of unions of all such intervals.

Reviewer #4: This is a review of a "Mathematical framework for place coding in the auditory system" by A. Reyes. This is a resubmission, but it my first time reading this manuscript (I did not review the 1st submission).

The author outlines a novel way to describe neural encoding, specifically a place code of sound frequency & intensity, using formal language of topology and set theory. The development of the theory is thoughtful and thought-provoking. Implications of the theory are pursued, specifically: how tones and tone combinations would be represented as spatial patterns of activation under the rules stipulated by the mathematical formulation. These outcomes of the formal model are then compared to simulations of an auditory cortex-inspired model, adapted from the author's previous work.

*** General comments ***

I enjoyed the formal mathematical perspective and appreciate that that author has attempted to tackle a fundamental question of neural representation in a novel way. It seems to me that there are a number of limitations of the approach that make me question whether it can be usefully applied to the auditory system. I expect the author can address these, and I think that readers of PLoS CompBio will enjoy reading this creative theoretical work. Major concerns are (some of these are related detailed comments below):

* this framework explicitly leaves out all temporal information of inputs and neural responses, and yet we know the auditory system is highly specialized to extract temporal information from inputs. It would be helpful for the author to expand on this in the Introduction (see some detailed comments below)

* this framework does not account for nonlinearities that are well-known to arise for even the most basic combinations of tones (some examples: perceptual phenomena such as combination tones and missing fundamental; two-tone suppression at the the cochlear level), and also does not account for complications of phase (as raised by Reviewer 2 in the previous round). The author should discuss how the model should be interpreted in light of these phenomena

* what is the utility (or anticipated utility) of the mathematical formalism & topological machinery introduced? It seems that that the model is defined by set operations (union, exclusion) on intervals. These are topics that could be familiar and intuitive to non-mathematically-sophisticated readers (and thus the model could be presented in a more mathematically "gentle" way, without much reference to topological terms). Of course, if the topological formulation leads to insights gained, for example, from "deeper" theorems or tools of topology, then the approach would be justified, but I don't see substantial uses of the topological approach in this manuscript. There is also mention of more esoteric algebraic structures in the supplemental material (monoid, magma) but no discussion of the significance of these structures in the current study.

* The formal model is compared to a spiking model. It is not stated clearly what this model represents. I take it to be a model for auditory cortex (based on the author's previous work, and the mention of "auditory cortical circuits" on line 244). The author should describe the rationale for this model. In particular, could the author expand on this choice as opposed to, say, comparing the formal predictions to activation patterns of auditory nerve simulations. Several well-known auditory nerve models are freely-available (see e.g. https://github.com/mrkrd/cochlea), the elements of tonotopy and place code are already established at this stage of auditory processing, and coding of the "simple" stimuli considered here (tones, tone combinations) would likely begin in the cochlea/ANF (whereas auditory cortex may be engaged in encoding "higher-level" auditory features).

*** Detailed comments ***

Line 8 (abstract): I think I was confused by the mention of the bijective mapping in the abstract (and also line 128). From this statement, I expected to see stronger results about what this framework would imply for decoding. But, as is discussed (line 146, also Fig 3) there is ambiguity in decoding. I would suggest some language to clarify that the bijection is on single tones, but not combinations of tones, or something similar?

Lines 11-12 (abstract): suggest removing "predicted", and just say "outcomes of these operations" or "resulting outcomes of these operations" [echoing some of Referee 2 and Referee 3's concerns from previous round]

Lines 40-45: Couldn't a counterargument be made here that acoustic information can possibly be carried by first-spike latency (Heil1997, Bizley et al 2010, etc)?

More generally, it seems that some discussion should be given to the fact that there are areas of the auditory system appear highly-specialized for temporal features of sounds and temporal precision in synaptic transmission, so a "place only" code (all temporal information ignored) may be an extreme way to view encoding in the auditory system.

Line 44-46: agree that "temporal or ‘volley’ schemes [are] difficult to implement at the level of cortex", but an alternate view is that temporal or volley information is used at some earlier stage and transformed for use by the auditory cortex, so AC does not have to implement a place code for tones. Does this matter for interpretation of the model?

Line 54-56. Could add: it is also a challenge to understand how place codes can work when inputs (sounds) are dynamic and temporally complex. [but then should acknowledge that temporal dynamics are not considered in this study]

General comment on Intro: would be helpful to be more explicit that the author's definition of "place code" means place ONLY, no temporal information of any kind considered in inputs or neural responses. and then comment on the meaningfulness & limitations of this approach for understanding the auditory system.

Line 62 - "Math model" - change to mathematical

Eq. 3: The statements in lines 504-507 should also be included near here to clarify that f and p are not defined on a logarithmic scale, as might be expected

Line 129-131 - please provide reference for statement that E and I receptive fields differ. this seems to refer to the question of "co-tuning" that has been considered by the author before, more context/explanation could be helpful here.

Line 136: closure is said to be "important". Not clear to me on what grounds. Important in terms of the ensuring the model is a sensible description of neural processes? Or important because of the author's goal to adhere to formal mathematical requirements?

Line 174: suggest a more specific statement: "because the set union operation has no inverse" [instead of "because there is no inverse"]

Fig 7: Could dashed lines for theory predictions be included here, as was done in Fig. 6, or are they too distinct?

Line 322: typo: "applied [to] a well-known"

"Relation to Difference Limens" section. This is indeed an interesting prediction based on how the model is constructed, but doesn't this prediction require the equivalence classes (partitions of acoustic and neural space) to be very "rigid"... sensitive to infinitesimally small changes that cross the boundaries of intervals. This would seem to contradict the idea (line 115) that this construction is "loose". Indeed I would agree that it more reasonable to think of the partition of acoustic and neural space as more of qualitative/approximate way of thinking about how the place code construction.

Line 355 style: fDL could be $f_{DL}$ (subscript)

Line 371: "overlapping *non-empty* intervals" [so inequality strict}

Line 375: FRA defined in supplemental but not defined in text

Line 376: typo: "equal to [the] larger of two"

Line 394-397. Multiplication not right distributive. Can this be expanded on? For instance, are there certain local connectivity circuits/motifs for which this approach is more or less relevant

Line 401-403: "response of neurons to pure tones and white noise differ". Can something more specific be said here as to how the model is "in line" with the data. The results as presented did not specifically consider the case of broadband (noise) stimuli.

Line 404: typo. "it [is] not trivially"

Line 406 - 413. "Transmission of acoustic info" section could be clarified. The terms homeomorphism & homomorphism should be defined for non-specialists.

the phrase "would necessitate that the topology of the source and target brain regions be the same" is loose here... "the same" in a formal sense that there is a homeomorphism between the two? or "the same" in some way that can be interpreted in terms of the source space and neural space.

Lines 412-413 are also vague as they do not specify what constraints the author has in mind.

GENERAL: Nonlinearities such as missing fundamental, combination tones

Line 414. Algebra of loudness summations. This could be presented as a result (and moved to Results).

Line 479: Could say tonotopy starts in the cochlea (von Bekesy) and auditory nerve, i.e. prior to auditory brainstem.

**Have all data underlying the figures and results presented in the manuscript been provided?**

Reviewer #1: Yes

PLOS authors have the option to publish the peer review history of their article (what does this mean?). If published, this will include your full peer review and any attached files.

Reviewer #1: No

Reviewer #4: No

**Have the authors made all data and (if applicable) computational code underlying the findings in their manuscript fully available?**

Reviewer #4: **No: **I may have missed it, but I don't see any information regarding availability or distribution of code used for network simulations.
---

## [Decision Letter · Decision Letter 2]

22 Jun 2021

Dear Alex, I think you are there, but I would appreciate if you could address the last comments of Reviewer 4.

Best,

Lyle

Dear Dr. Reyes,

Thank you very much for submitting your manuscript "Mathematical framework for place coding in the auditory system" for consideration at PLOS Computational Biology. As with all papers reviewed by the journal, your manuscript was reviewed by members of the editorial board and by several independent reviewers. The reviewers appreciated the attention to an important topic. Based on the reviews, we are likely to accept this manuscript for publication, providing that you modify the manuscript according to the review recommendations.

Sincerely,

Lyle J. Graham

Deputy Editor

PLOS Computational Biology

[LINK]

Reviewer's Responses to Questions

**Comments to the Authors:**

Reviewer #1: The author addressed my comments.

Reviewer #4: The author has revised the manuscript and addressed my concerns. In particular, several of the additions (including lines 46, 61, lines 475-488) usefully define the scope and limitations of the study. The technical approach is clearly developed in the supporting documents. This material will be much appreciated by readers who want to dive deeper into the interesting and novel results presented in the paper.

The paper includes simulation results from a computational model. I do not see it stated if/where that code is available. Apologies if I missed it.

Two comments that I expect the author can address with some further minor revisions are:

* lines 113-115

"Theoretically, frequencies and pressures are unbounded and can take an uncountable number of values but under physiological conditions, the audible range is likely bounded by minimum and maximum values and is finite."

A previous reviewer flagged an earlier version of this passage, noting that finite intervals can contain uncountably many elements (for real numbers, e.g.). I think this wording could still be adjusted to avoid confusion. One simple fix could be to delete "and can take an uncountable number of values"

However, my sense is that the author is purposeful in using the word "uncountable" in order to draw a contrast with the fact that the model has finitely-many elements in acoustic space. In this case, maybe say something like

Theoretically, frequencies and pressures are unbounded and can take an uncountable number of values but under physiological conditions, the audible range of tones is likely bounded by minimum and maximum values and can be partitioned (due to JNDs / difference limens) into finitely-many discriminably-different frequencies and amplitudes

...

or worded differently, to the author's liking

* First spike latency comment. Please note: I am not an expert on this particular issue of first spike latency and have no connection to the relevant work. I am following up on this point simply because I am trying to think through the correct interpretation of the model.

In the manuscript the author writes:

[lines 40-44]: This is of some significance because in this short time interval, neurons can fire only bursts of 1-2 action potentials [14, 15], indicating that neurons essentially function as binary units. Therefore, it seems likely that neither frequency nor intensity can be encoded via the firing rate of individual cells since the dynamic range would be severely limited. "

In the letter to reviewers the author wrote:

"the Bizley study showed that first spike latency can be used to discriminate between higher and lower frequency stimuli"

so perhaps it would be appropriate to to modify the phrase "it seems likely that neither frequency nor intensity can be encoded via the firing rate of individual cells" could be modified to say something like.... single spikes may carry some information -- for instance first-spike spike latency may carry some frequency info -- but this frequency info is ambiguous (not invariant to sound pressure level) and likely restricted to narrow ranges of sound pressure level, so for simplicity we exclude this possible source of info in our construction of a place-code model

**Have the authors made all data and (if applicable) computational code underlying the findings in their manuscript fully available?**

Reviewer #1: Yes

Reviewer #4: **No: **The paper includes simulation results from a computational model. I do not see it stated if/where that code is available. Apologies if I missed it.

PLOS authors have the option to publish the peer review history of their article (what does this mean?). If published, this will include your full peer review and any attached files.

Reviewer #1: No

Reviewer #4: No

Figure Files:

Data Requirements:

Reproducibility:

References:

---

## [Editor Report · Decision Letter 3]

6 Jul 2021

Dear Alex, I'm very glad this is in, and thanks for your responses throughout the process. Best, Lyle

Dear Dr. Reyes,

We are pleased to inform you that your manuscript 'Mathematical framework for place coding in the auditory system' has been provisionally accepted for publication in PLOS Computational Biology.

Best regards,

Lyle Graham

Deputy Editor

PLOS Computational Biology

---

## [Editor Report · Acceptance letter]

28 Jul 2021

PCOMPBIOL-D-20-01254R3 

Mathematical framework for place coding in the auditory system

Dear Dr Reyes,

I am pleased to inform you that your manuscript has been formally accepted for publication in PLOS Computational Biology. Your manuscript is now with our production department and you will be notified of the publication date in due course.

With kind regards,

Zsofi Zombor
